# Tumor Stimulus-Responsive Biodegradable Diblock Copolymer Conjugates as Efficient Anti-Cancer Nanomedicines

**DOI:** 10.3390/jpm12050698

**Published:** 2022-04-27

**Authors:** Vladimír Šubr, Robert Pola, Shanghui Gao, Rayhanul Islam, Takuma Hirata, Daiki Miyake, Kousuke Koshino, Jian-Rong Zhou, Kazumi Yokomizo, Jun Fang, Tomáš Etrych

**Affiliations:** 1Institute of Macromolecular Chemistry, Czech Academy of Sciences, Heyrovského nám. 2, 16206 Prague, Czech Republic; subr@imc.cas.cz (V.Š.); pola@imc.cas.cz (R.P.); 2Faculty of Pharmaceutical Sciences, Sojo University, Ikeda 4-22-1, Nishi-ku, Kumamoto 860-0082, Japan; gaoshanghui94@gmail.com (S.G.); rayhanulislam88@gmail.com (R.I.); g1851110@m.sojo-u.ac.jp (T.H.); g1851129@m.sojo-u.ac.jp (D.M.); g1851054@m.sojo-u.ac.jp (K.K.); zhoujr@ph.sojo-u.ac.jp (J.-R.Z.); yoko0514@ph.sojo-u.ac.jp (K.Y.)

**Keywords:** pirarubicin, drug delivery, HPMA conjugate, diblock conjugate, anticancer

## Abstract

Biodegradable nanomedicines are widely studied as candidates for the effective treatment of various cancerous diseases. Here, we present the design, synthesis and evaluation of biodegradable polymer-based nanomedicines tailored for tumor-associated stimuli-sensitive drug release and polymer system degradation. Diblock polymer systems were developed, which enabled the release of the carrier drug, pirarubicin, via a pH-sensitive spacer allowing for the restoration of the drug cytotoxicity solely in the tumor tissue. Moreover, the tailored design enables the matrix-metalloproteinases- or reduction-driven degradation of the polymer system into the polymer chains excretable from the body by glomerular filtration. Diblock nanomedicines take advantage of an enhanced EPR effect during the initial phase of nanomedicine pharmacokinetics and should be easily removed from the body after tumor microenvironment-associated biodegradation after fulfilling their role as a drug carrier. In parallel with the similar release profiles of diblock nanomedicine to linear polymer conjugates, these diblock polymer conjugates showed a comparable in vitro cytotoxicity, intracellular uptake, and intratumor penetration properties. More importantly, the diblock nanomedicines showed a remarkable in vivo anti-tumor efficacy, which was far more superior than conventional linear polymer conjugates. These findings suggested the advanced potential of diblock polymer conjugates for anticancer polymer therapeutics.

## 1. Introduction

Many routinely used anticancer drugs suffer from poor pharmacokinetics and inappropriate biodistribution. Due to their low molecular weight, drugs are generally rapidly removed from the circulation, and thus not only accumulate at the pathological site, but also cause toxicity in various healthy tissues [1]. To achieve the correct pharmacokinetics of anti-cancer drugs and their accumulation at the target site, many drug delivery systems have been proposed over the years.

Nanomedicines are submicrometer-sized carrier materials, which intend to improve the biodistribution of systemically administered anticancer drugs. By delivering pharmacologically active agents more selectively to pathological sites, and/or by guiding them away from potentially endangered healthy tissues, nanomedicine formulations aim to improve the balance between the efficacy and the toxicity of systemic therapeutic interventions [2,3,4,5]. Nanomedicines refer to the application of therapeutics based on nanotechnologies not only for the treatment of cancer but also for imaging agents for diagnostics. Prototypic examples of nanomedicine formulations are liposomes [6], polymer conjugates [5,7], micelles [8,9], polymersomes [10,11], dendrimers [12], nanoparticles [13] and antibodies [14].

Increasing the size of systemically administered nanomedicines, with a diameter of at least 8–10 nm and a molecular weight of 50,000 g/mol, close to the renal threshold of the kidneys leads to the prolongation of their circulating half-life in the bloodstream and increased accumulation in the target tumor tissue. Importantly, some nanomedicines, e.g., liposomes [6], nanoparticles [13] and polymer–drug conjugates [5,7,15], were found to be less therapeutically active due to the low stability and leakage of the drug in systemic circulation or due to insufficient release of drugs at target sites [15,16]. Nanomedicines are known to have a slower intracellular uptake and a somewhat reduced cytotoxicity [17]. Therefore, the right stimuli activation of the carried drug is the key point of free drug release from nanomedicines at the tumor tissue to observe the extended therapeutic activity [18]. Drug delivery systems that combine anticancer drugs with water-soluble polymer carriers are highly promising and offer an exploration of safer cancer therapies that are more specific and efficient. The versatility of the polymer carrier structure and its potential for modification makes it possible to tune the properties of the polymer−drug conjugate to the needs of specific treatments.

To prolong the circulating half-life in the bloodstream, and thus increase accumulation in the targeted tissue, high-molecular-weight (HMW) water-soluble polymer conjugates, which contain degradable bonds that enable the degradation of HMW conjugates to parts excretable by the kidneys, and have a molecular weight below 50,000 g/mol, were developed and synthesized. Such degradable bonds can be hydrolytically, enzymatically or reductively degradable, e.g., HMW diblock copolymers, containing a Gly-Phe-Leu-Gly-Lys-Gly-Phe-Leu-Gly degradable bond by protease cathepsin B [19], were synthesized.

Matrix metalloproteinases (MMPs) are zinc-dependent endopeptidases specific to extracellular matrix (ECM) remodeling in various physiological and pathological processes by cleaving ECM proteins such as fibronectin, collagen, laminin and elastin [20]. Their activity is dependent on metal ions, such as zinc or calcium [21]. They also play important roles in other physiological processes, such as organ morphogenesis, embryonic development, apoptosis, wound healing and angiogenesis [22,23]. MMPs are expressed at low levels in normal physiological conditions. They are overexpressed in a number of diseases such as various types of cancer, central nervous system disorders, cardiovascular diseases, neurodegenerative diseases and arthritis [24,25,26,27]. The HMW diblock copolymer carrier containing a degradable disulfide bond between the polymer blocks showed a rapid degradation in the presence of the cytoplasmatic concentration of glutathione [28].

Copolymers of *N*-(2-hydroxypropyl)methacrylamide (HPMA) rank among the most frequently studied water-soluble drug carriers. These polymers are known as highly non-fouling polymers; importantly, no soft or hard protein corona around the HPMA copolymers was observed [29]. A large body of information has been collected concerning their in vitro activity and mechanisms of action as well as their in vivo activity, mainly in mice [30] but also in humans [31,32,33]. Recently, we described pHPMA polymer nanomedicines carrying pirarubicin (THP), bound to the polymer chain via a pH-sensitive hydrazone bond known to be cleavable in the mild acidic environment of the tumor tissue [34,35]. Moreover, the hydrazone bond is known to be responsive to any decrease in the pH from the neutral values, and thus the release could already be observed in the extracellular space of the tumor tissue and could improve the cytotoxic profile of the whole nanomedicine [36,37,38,39]. Indeed, the therapeutic efficacy could be improved by the formation of the HMW systems based on the diblock, grafted or star-shaped polymer systems [38,39,40].

Here, we present the synthesis and biological evaluation of the novel diblock pHPMA nanosystems intended for the enhanced tumor accumulation and tumor pH-associated drug release and re-activation. The diblock polymer structure degradable by the matrix metalloproteinases or by reductive degradation was designed and synthesized by employing classic and controlled radical polymerization. The results of this biological evaluation provide a deep insight into the structure-to-efficacy relationship and prove the increased efficacy of the biodegradable diblock polymer nanomedicines.

## 2. Materials and Methods

### 2.1. Materials

*N*,*N*′-dicyclohexylcarbodiimide (DCC) and *N*-ethyl-*N*′-(3-dimethylaminopropyl)carbodiimide hydrochloride (EDC); *N*,*N*′-diisopropylcarbodiimide (DIC), triisopropylsilane (TIPS), *N*,*N*-dimethylformamide (DMF), *N*,*N*-dimethyl-4-aminopyridine (DMAP), dichloromethane (DCM), methanol, ethyl acetate, 2,2′-azobisisobutyronitrile (AIBN), 4,4′-azobis(4-cyanopentanoic acid) (AACVA), 1-aminopropan-2-ol, methacryloyl chloride, methyl-6-aminohexanoate hydrochloride (ah), *tert*-butyl carbazate, and *N*,*N*-diisopropylethylamine (DIPEA); *N*-ethyldiisopropylamine (EDPA), pentafluoro phenol, dithiothreitol (DTT), and ethanethiol; carbon disulfide; sodium hydride (60% dispersion in mineral oil), cystamine dihydrochloride, dimethyl sulfoxide (DMSO), *tert*-butyl alcohol (t.BuOH), silica gel 60, and tris(2-carboxyethyl)phosphine hydrochloride (TCEP) were purchased from Sigma-Aldrich (Prague, Czech Republic). Pirarubicin (THP) was purchased from Meiji Seika (Tokyo, Japan), and 2,4,6-trinitrobenzene-1-sulfonic acid (TNBS) was purchased from Serva (Heidelberg, Germany). 2,2′-azobis(4-methoxy-2,4-dimethylvaleronitrile) (V-70) was purchased from FUJIFILM Wako Chemicals Europe GmbH. All other chemicals and solvents were of analytical grade. The solvents were dried and purified by conventional procedures and distilled before use.

9-fluorenylmethoxycarbonyl-amino acids (Fmoc-AA), *N*,*N*-dimethylformamide (DMF), ethyl cyanoglyoxylate-2-oxime (Oxyma), Tenta Gel R RAM resin, piperidine (Pip), trifluoroacetic acid (TFA) were purchased from Iris Biotech, GmbH, Marktredwitz, Germany.

RPMI 1640 medium, Dulbecco’s modified eagle medium (DMEM), and other reagents and solvents were purchased from Wako Pure Chemical (Osaka, Japan). Fetal calf serum was obtained from GIBCO (Grand Island, NY, USA). 3-(4,5-dimethyl-2-thiazolyl)-2,5-diphenyl-2H-tetrazolium bromide (MTT) was purchased from Dojindo Chemical Laboratories (Kumamoto, Japan).

### 2.2. Synthesis of Monomers, Oligopeptide and Chain Transfer Agents

*N*-(2-Hydroxypropyl)methacrylamide (HPMA) was synthesized by reaction of methacryloyl chloride with 1-aminopropan-2-ol, as previously described, using Na_2_CO_3_ as a base [41]. *N*-Methacrylamidohexanoylhydrazine (Ma-Ahx-NHNH_2_) was prepared as previously described [39]. *tert*-Butyl-*N*-[6-(2-methylprop-2-enoylamino) hexanoylamino]carbamate (Ma-Ahx-NHNH-Boc) was prepared in a two-step synthesis, as previously described [40].

4-Cyano-4-(ethylthiocarbonothioylthio)pentanoic acid was synthesized as described earlier [42].

The purity of the monomers and chain transfer agents was examined by HPLC system (Shimadzu, Japan) equipped with a reverse-phase column (Chromolith Performance RP-18e, 100 mm × 4.6 mm) using elution with water–acetonitrile (gradient 0–100% acetonitrile) utilizing UV-VIS photodiode array detection (Shimadzu SPD-M10A vp) (220 nm–250 nm).

#### 2.2.1. Synthesis of PVGLIGK Peptide

Peptide sequence PVGLIGK (see Figure 1) was assembled on solid phase (TentaGel resin R RAM) using automatic Liberty Blue microwave peptide synthesizer (CEM, Matthews, NC, USA), starting from the C-terminus using standard Fmoc procedures with the consecutive addition of the *N*-Fmoc-protected amino acid derivative (2.5 equiv.), DIC (2.5 equiv.) as an activator and Oxyma (2.5 equiv.) as an activator base, all in DMF. After attaching the *N*-Fmoc-Pro-OH and removing the Fmoc, the final peptide was cleaved from the resin using a mixture of 95 vol% TFA, 2.5 vol% TIPS and 2.5 vol% water for 1 h. The resin was removed by filtration, the filtrate was concentrated under reduced pressure, and the crude peptide was isolated by precipitation to cold diethyl ether followed by filtration. Purity and identity of the prepared peptides was determined by HPLC and MALDI-TOF MS (calculated 682.4, found 683.5 M+H).

#### 2.2.2. Synthesis of Pentafluoro Phenyl Functional Chain Transfer Agent (2,3,4,5,6-Pentafluorophenyl) 4-Cyano-4-Ethylsulfanylcarbothioylsulfanyl-Pentanoate (CTA-PFP)

4-Cyano-4-(ethylthiocarbonothioylthio)pentanoic acid (0.387 g, 1.47 mmol) and pentafluoro phenol (0.28 g, 1.54 mmol) were dissolved in 10 mL of dichloromethane (DCM) and *N*-ethyl-*N*′-(3-dimethylaminopropyl)carbodiimide hydrochloride (0.41 g, 2.16 mmol) was added to the solution followed by a catalytic amount of DMAP. The reaction mixture was stirred at a laboratory temperature for 2 h and extracted with distilled water (2 × 10 mL), before the organic layer was dried with Na_2_SO_4_ and the DCM evaporated. The remaining oily residue was diluted with a mixture of hexane:ethyl acetate (1:2; 5 mL) and purified by silica gel chromatography using hexane:ethyl acetate (1:2). The eluent was evaporated and the resulting yellow oily product was crystallized in a refrigerator. The final yield was 0.48 g (76%). HPLC on Chromolith RP18e (100 mm × 4.6 mm) gave a single peak with retention time 12.6 min using gradient of mobile phase 0–100% H_2_O-Acetonitril with 0.1% of TFA.

#### 2.2.3. Synthesis of 4-Cyano-N-[2-[2-[(4-Cyano-4-Ethylsulfanylcarbothioylsulfanyl-Pentanoyl)Amino]ethyldisulfanyl]ethyl]-4-Ethylsulfanylcarbothioylsulfanyl-Pentanamide (CTA-S-S-CTA)

CTA-PFP (0.48 g, 1.12 mmol) was dissolved in DMF (3.0 mL) and cysteamine dihydrochloride (0.126 g, 0.56 mmol) followed by addition of DIPEA (195 µL, 1.12 mmol). Reaction mixture was stirred for 4 h at room temperature. DMF was evaporated in vacuo, reaction mixture was dissolved in ethyl acetate, and CTA-S-S-CTA was purified by silica gel chromatography using ethyl acetate. The eluent was evaporated, and the resulting yellow oily product was stored in a refrigerator. The final yield was 0.47 g (65%). HPLC on Chromolith RP18e (100 mm × 4.6 mm) gave single peak with retention time 12.2 min using gradient of mobile phase 0–100% H_2_O-Acetonitril with 0.1% of TFA. ESI MS (M+Na 664.92), see Figure 1.

### 2.3. Synthesis of the Polymer Precursors and Polymer Conjugates

Linear polymer conjugate L-THP, poly(HPMA-co-Ma-Ahx-NHN=THP), bearing drug bound via pH-sensitive hydrazone bond was synthesized as described elsewhere [35]. The reductively degradable diblock conjugate D-ss-THP, poly(HPMA-co-Ma-Ahx-NHN=THP)-SS-poly(HPMA-co-Ma-Ahx-NHN=THP), conjugate containing THP attached via a hydrazone bond was prepared by THP conjugation with the diblock polymer precursor, as recently described [43]. The structures of polymer conjugates, L-THP and D-ss-THP, D-_PVGLIGK_-THP and r-D-ss-THP are shown in Figure 2 and their characteristics are summarized in Table 1.

Polymer precursor D-PVGLIK-Hy, poly(HPMA-co-Ma-Ahx–NHNH–Boc)-PVGLIGK-poly(HPMA-co-Ma-Ahx–NHNH–Boc), linear diblock copolymer containing enzymatically, matrix-metalloproteinase-sensitive, degradable sequence, was prepared by the reaction of thiazolidine-2-thione TT groups located on main-chain end of polymer precursor L-TT with PVGLIGK oligopeptide. The polymer precursor L-TT was prepared by the radical solution copolymerization of HPMA and Ma-Ahx-NHNH-Boc as described [38]. An example of the synthesis is as follows: polymer L-TT (250 mg, 14.6 μmol TT groups) was dissolved in 3 mL of DMF, and DIPEA (2.5 μL) was added under stirring. The solution of PVGLIGK oligopeptide (5 mg, 7.3 μmol) in 1 mL DMF was continuously added into the stirred L-TT solution within 10 min. After 1 h, 1-aminopropan-2-ol (2 μL) was added to block unreacted TT groups. Polymer precursor D-PVGLIK-Hy was separated by precipitation into ethyl acetate with following deprotection of Boc groups in concentrated TFA.

HPMA copolymer r-D-ss-Hy and raft-poly(HPMA-co-Ma-Ahx–NHNH–Boc)-SS-poly(HPMA-co-Ma-Ahx–NHNH–Boc), (see Figure 2), were prepared by RAFT copolymerization of HPMA and Ma-Ahx–NHNH–Boc using 2,2′-azobis(4-methoxy-2,4-dimethylvaleronitrile) (V-70) as an initiator and CTA-S-S-CTA as a CTA in molar ratios of monomer:CTA:initiator 700:1:1. The molar ratio of HPMA to Ma-Ahx–NHNH–Boc in the reaction mixture was 90:10. An example of the synthesis of r-D-ss-Hy is as follows: HPMA (1.0 g, 6.98 mmol) and Ma-Ahx–NHNH–Boc (0.243 g, 0.77 mmol) were dissolved in 11.1 mL of tert-butyl alcohol and mixed in a polymerization ampule with CTA-S-S-CTA (7.13 mg, 11.08 μmol) and initiator V-70 (3.42 mg, 11.08 μmol) and then dissolved in 0.745 mL of anhydrous DMA. The copolymerization mixture was bubbled with argon for 10 min and sealed. Copolymerization was carried out at 40 °C for 16 h. The copolymer was isolated by precipitation in a mixture of acetone and diethyl ether (2:1), filtered off and dried under a vacuum. The yield was 0.70 g (56%). The average molecular weight Mw was 77,000 mol/g and dispersity Ð was 1.05. Boc protecting groups were removed by dissolving the polymer precursor in TFA, followed by precipitation in diethyl ether, filtration off, washed with diethyl ether and dried in vacuo. The content of hydrazide groups 6.5 mol% was determined using TNBSA method [36].

Diblock polymer conjugates D-PVGLIK-THP, poly(HPMA-co-Ma-Ahx–NHN=THP)-PVGLIGK-poly(HPMA-co-Ma-Ahx–NHN=THP), r-D-ss-THP, and raft-poly(HPMA-co-Ma-Ahx–NHN=THP)-SS-poly(HPMA-co-Ma-Ahx–NHN=THP) were obtained by the reaction of the respective polymer precursor with THP in dried methanol for r-D-ss-THP as follows: Polymer precursor r-D-ss-Hy (585 mg) and pirarubicin (50 mg) were dissolved in 3.5 mL of dried methanol, and acetic acid (0.05 mL) was added. Reaction mixture was stirred for 24 h at room temperature in the dark, diluted with 15 mL of methanol, and polymer conjugate was purified on Sephadex LH-20 in methanol. Methanol was evaporated, and oily residue was diluted to 15 wt% solution with methanol and precipitated into ethyl acetate. The yield of the polymer conjugate r-D-ss-THP was 500 mg. The content of pirarubicin was spectrophotometrically determined and was 9.03 wt%. The characteristics of the polymer precursors and conjugates are shown in Table 1, and the structures of polymer conjugates are shown in Figure 2.

### 2.4. Characterization of the Polymer Precursors and Conjugates

The average molecular weight *M*_w_ and the dispersity Đ of the polymers were calculated using Shimadzu HPLC system equipped with multiangle light scattering DAWN 8, and refractive index Optilab rEX (Wyatt Co., Santa Barbara, CA, USA) detectors. The HPLC was equipped with TSKgel G3000SWXL or TSKgel G4000SWXL column (300 mm × 7.8 mm, 5 μm) with mobile phase consisting of 20% 0.3 M acetate buffer (pH 6.5) and 80% methanol at flow rate of 0.5 mL min^−1^. The dynamic light scattering (DLS) of the polymer (1 wt% solution) in phosphate buffer of pH 7.4 was measured at a 173° scattering angle on a Zetasizer ZEN3600 instrument (Malvern, UK).

The content of the hydrazide groups in the polymer precursor was determined by a modified TNBS assay, as described previously, using ε_500nm_ = 17,200 L mol^−1^ cm^−1^ for estimation of -NHNH_2_ group [36].

The total content of THP in the polymer conjugates was determined spectrophotometrically on a Helios α spectrophotometer, using ε_488nm_ = 11,500 L mol^−1^ cm^−1^ in water.

### 2.5. In Vitro Drug Release from the Polymer Conjugates

All the studied polymer conjugates were dissolved at 5 mg/mL in PBS and sealed in dialysis bag (1 mL), then merged in sodium phosphate buffers of different pH (pH 5.5, 6.5 and 7.4, 10 mL) incubated at 37 °C with shaking. In predetermined times, 200 µL of solution from the dialysis bag was collected, and the absorbance at 488 nm of THP was measured by a plate reader. A standard curve of free THP was used to determine the amount of released free THP.

### 2.6. Degradation of Diblock Copolymers

In vitro enzymatic degradation of PVGLIGK spacer containing diblock polymer and conjugate was performed in presence of the MMP-2 and MMP-9. The enzymatic degradation was carried out in Dulbecco’s phosphate-buffered saline (pH 7.5) at 37 °C at concentration 4 mg mL^−1^. The degradation was initiated by the addition of 100 µL MMP-2 or MMP-9 stock solution (2 × 10^−7^ M) to the 0.5 mL solution of diblock copolymer. The stock solution of the MMP-2 or MMP-9 was prior to the addition of polymer solution activated with 10 µL of 100 mM APMA solution in DMSO at 37 °C for 16 h. At selected time intervals, an aliquot was withdrawn, and the *M*_w_ of polymer degradation products was determined by size exclusion chromatography as described above with DAWN 8 and Optilab rEX detectors. All experiments were performed in triplicate.

Model reduction in the disulfides of polymer conjugates, r-D-ss-Hy and r-D-ss-THP, in water was carried out in the presence TCEP. Conjugate r-D-ss-Hy or r-D-ss-THP (5 mg) was dissolved in PBS buffer (1 mL), and 10 µL of TCEP (10 mg/1 mL water) was added. After 1 h, the change in the GPC profile of polymer conjugate before and after disulfide reduction was compared by size exclusion chromatography with DAWN 8 and Optilab rEX detectors (See Appendix A).

### 2.7. In Vitro Cytotoxicity Assay

Mouse colon cancer C26 cells (Riken Cell Bank, Tokyo, Japan) were cultured in RPMI-1640 medium containing 10% fetal bovine serum (FBS) and 100 units/mL penicillin-streptomycin under a humidified atmosphere of 5% CO_2_ at 37 °C. For cytotoxicity assay, the cells were seeded in 96-well plate (5000 cells/well). After overnight preincubation, P-THPs (THP equivalent) were added into the cells at indicated concentrations. After further 48 h incubation, cell viability was measured by MTT assay.

### 2.8. Intracellular Uptake of Conjugates in C26 Cells

C26 cells were seeded in 12-well plate (10^5^ cells/well), and after overnight preincubation, different P-THPs were added at concentration of 30 µg/mL (THP equivalent). After indicated time, media were removed, and the cells were washed by PBS for 3 times. The cells were then collected after trypsin treatment, and the collected cells were subjected to sonication (30 s), after which 10 M HCl was added to each tube to the final concentration of 1 M. After 1 h incubation in room temperature, the solutions were centrifuged (12,000 rpm, 5 min), and the obtained supernatants were used for FL measurement using a plate reader (Ex488/Em590).

In a separate experiment, the cells cultured in the plates were subjected to FL microscopical examination (Ex488/Em590) using a fluorescence microscope (Keyence BZ-X700, Keyence Corporation, Osaka, Japan).

### 2.9. Spheroid Assay for the Penetration and Uptake of Different P-THPs

C26 cells (2 × 10^5^) were seeded in 14 cm^2^ ultra-low attachment cell dishes (Corning Inc., Corning, NY, USA) and cultured for 6 days to form cell spheroids. Then, the spheroids were transferred into 9.6 cm^2^ glass-bottom culture dishes, and P-THPs were added in a final concentration of 30 μg/mL (THP equivalent). At indicated time, the drugs that penetrated into the spheroids and/or internalized into the cells were visualized using confocal laser fluorescence microscopy, with the excitation wavelength at 488 nm and emission wavelength at 570−640 nm (Nikon TE2000U, Nikon Solutions Co., Ltd., Tokyo, Japan).

### 2.10. In Vivo Pharmacokinetics of P-THPs

Healthy male ddY mice, aged 6 weeks and from SLC Inc. (Shizuoka, Japan), were used to investigate the in vivo pharmacokinetics of P-THPs. All animals were maintained under standard conditions and fed water and murine chow ad libitum. All animal experiments were approved by the Animal Ethics Committees of Sojo University (no. 2021-P-024, approved on 1 April 2021) and were carried out according to the Guidelines of the Laboratory Protocol of Animal Handling, Sojo University.

P-THPs were injected i.v. into ddY mice. After the indicated time, 50–100 µL of blood was collected from the tail vein and placed in Eppendorf tube rinsed with heparin, and plasma was obtained by centrifugation (2500× *g*, 4 °C, 15 min). To 50 µL of serum, 450 µL of PBS was added, to which 50 µL of 10 M HCl (final concentration of 1 M) was added, then subjected to sonication for 30 s. The homogenate was then incubated at 50 °C for 1 h to hydrolyze THP derivatives. Then, the aglycon was extracted by adding 1 mL of chloroform. After vigorous vortex and centrifugation (12,000 rpm, 5 min), 0.5 mL of the chloroform phase was shifted to other tube and then evaporated to dryness. The obtained pellet was dissolved in the mobile phase of HPLC and analyzed by Hitachi Chromaster HPLC system (Hitachi High Tech. Sci., Tokyo, Japan) with a 5410 UV detector at 488 nm. The column was Capcell Pak SCX UG column (4.6 mm × 250 mm) (Osaka Soda Co., Ltd., Osaka, Japan), and the column temperature was maintained at 25 °C. The mobile phase consisted of 33% acetonitrile and 67% 0.1 M sodium acetate buffer (pH 5.0) at a flow rate of 1.2 mL/min.

### 2.11. In Vivo Antitumor Effect

Mouse sarcoma S180 solid-tumor model was used in this study, in which S180 cells (2 × 10^6^ cells) that had been grown in peritoneal cavity of ddY mice as ascetic form were subcutaneously implanted (s.c.) in the dorsal skin of ddY mice. At 7–10 days after tumor inoculation, when the diameters of tumors reached approximately 8–10 mm, the mice received bolus injection of P-THPs (5 mg/kg, THP equivalent). During the study period, the width (W) and length (L) of the tumors, as well as the body weight of mice, were measured every 2–3 days, and tumor volume (mm^3^) was calculated as (W^2^ × L)/2.

## 3. Results

The overall efficacy of the nanomedicines is strongly connected with their ability to accumulate in the tumorous tissue to a significantly higher extent. Here, we introduce diblock polymer-based nanomedicines, which are tailored to effective tumor accumulation, tumor-associated stimuli drug activation and the consequent degradation of the carrier system to ensure elimination from the body.

### 3.1. Synthesis of the Polymer Precursors

Two techniques of radical copolymerization, as well as free radical and RAFT polymerization, were employed for the synthesis of the polymer precursors. The diblock polymer precursors were prepared by two different strategies: (i) by combination of two linear polymers via biodegradable linkage, or (ii) via RAFT copolymerization using bifunctional CTA agent enabling the simultaneous growth of two linear polymers on the biodegradable core. Reductively or enzymatically degradable linkages were employed as suitable degradable parts ensuring biodegradation after the successful delivery of the cargo to the tumor tissue.

In the first approach, we synthesized a linear semitelechelic polymer precursor containing the protected hydrazide groups and main-chain end terminating amino-reactive TT group. The molecular weight was set to 30,000 g/mol to ensure the elimination of the linear polymers after diblock degradation. Either cysteamine or PVGLIGK oligopeptide were introduced as biodegradable linkages for the diblock formation. The diblock formation was controlled by the slow addition of the cysteamine or PVGLIGK into the stirred polymer solution. Both diblock precursors, D-_PVGLIGK_-Hy and D-ss-Hy, almost doubled *M*_w_ and increased hydrodynamic size. In the second approach, the bifunctional chain-transfer agent CTA-S-S-CTA was developed and applied to the controlled one-pot synthesis of diblock polymer in situ during controlled RAFT polymerization. Such an approach enabled the diblock polymer precursor to be easily synthesized with appropriate *M*_w_ and low dispersity. All polymer precursors contained sufficient amounts of hydrazide groups for the attachment of the drug molecules.

### 3.2. Synthesis of the Polymer–Drug Conjugates

All polymer precursors were employed in the synthesis of the polymer–THP conjugates, see Table 1. All polymer conjugates contained sufficient amounts of the THP, 9–10 wt%. Importantly, the sizes of the diblock polymer–THP conjugates were significantly higher than those of the control linear counterpart. The dispersity of the r-D-ss-THP was somewhat increased, in contrast to the respective polymer precursor, r-D-ss-Hy. We assume that the higher increase in the molecular weight and dispersity could be ascribed to the determination of *M*_w_ using multi-angle light scattering (MALS) detection. As the light of the laser used in the MALS analysis could also be absorbed by the carried THP molecule, the consequent fluorescence of THP partly increased the signal of the MALS diode detection. Such a fluorescence phenomenon therefore increased the response of the MALS detection and seemingly determined the increase in the molecular weight of the conjugate. Importantly, the sizes of all the diblock polymer–THP conjugates were approximately the same, almost 2-fold of the size of L-THP, proving the usability of samples for further biological experiments.

### 3.3. Release of Drugs from Polymer Conjugates

The stability and release behavior of THP from polymer conjugates was studied in buffered solution with three pHs modeling the blood stream conditions (pH 7.4), slightly acidic extracellular microenvironment of tumor cells (pH 6.5) and endosomal conditions of tumor cells (pH 5.5). For all the polymer conjugates, regardless of their inner polymer structure, a strong pH-responsive THP release behavior was found. The THP was slowly released at pH 7.4 mimicking the blood stream conditions to reach 20–30% of liberated THP after 24 h. On the contrary, 40–65% of THP was released in the same period of time at pH 6.5 and more than 70% at pH 5.5. The data confirmed the pH-responsive behavior of both the control linear polymer and novel diblock polymer–THP conjugates. The release of free THP from polymer conjugates is shown in Figure 3.

### 3.4. In Vitro Degradation of Diblock Conjugates

The degradability of the developed diblock polymer systems was determined either in the reductive conditions or presence of the MMP-2 and MMP-9 enzymes. The reductive degradability of D-ss-Hy and D-ss-THP was recently proved [43], showing the degradation of the diblock polymer system under the limit of the renal threshold in 2 h. Importantly, the diblock polymer system was stable in blood plasma for more than 48 h. Similarly, a reductive degradation was proven for diblocks r-D-ss-Hy and r-D-ss-THP prepared by the controlled RAFT polymerization. Here, the model incubation was carried out using presence of TCEP for 1 h, see Appendix A. Both deblocks, r-D-ss-Hy and r-D-ss-THP, regardless of the content of the THP, were degraded to polymer degradation products with a molecular weight close to the limit of the renal threshold.

Similarly, the diblock D-_PVGLIGK_-THP conjugate was studied for degradability using the MMP-2 and MMP-9 enzymes; see Figure 4. The diblock polymer conjugate was degraded within 24 to polymer degradation products with a molecular weight significantly under the limit of the renal threshold. Indeed, the diblock D-_PVGLIGK_-THP conjugate was stable during the incubation in plasma, and there was no change in molecular weight within 96 h of incubation, thus proving the stability of the diblock during the circulation in the body.

### 3.5. In Vitro Cytotoxicity of Diblock Conjugates

As shown in Figure 5, a dose-dependent cytotoxicity of L-THP was observed, which was consistent with our previous study using L-THP [35]. All diblocks exhibited similar cytotoxicity profiles to that of L-THP, with an IC_50_ of 0.09–0.15 ug/mL (THP equivalent).

### 3.6. Intracellular Uptake of Diblock Conjugates

When we examined the intracellular uptake of the P-THP conjugates, we found a time-dependent increase in the uptake of all P-THP conjugates, as confirmed by both fluorescence microscopy (Figure 6A) and a quantitative assay (Figure 6B). Among the three diblocks, r-D-ss-THP showed a relatively higher internalization than the others (Figure 6B); however, no statistically significant difference was observed. These findings are in parallel with the cytotoxicity assay as described in Figure 5.

### 3.7. Penetration and Uptake of P-THP Conjugates in C26 Tumor Cell Spheroid

We then examined the intratumor behaviors of P-THP conjugates by using C26 cell spheroids. In accordance with our previous study [44], L-THP penetrated into the spheroid in a time-dependent manner, which distributed almost a whole spheroid in 6 h (Figure 7). All diblocks exhibited similar behaviors; however, D-ss-THP and r-D-ss-THP showed a more rapid penetration than others (Figure 7).

### 3.8. In Vivo Pharmacokinetics of P-THP Conjugates

P-THP, as polymer conjugate drug exhibit prolonged circulation time compared to the native free THP (i.e., plasma t_1/2_ of >4 h vs. <1 h of free THP) [35]. We confirmed this result in this study, in which L-THP showed a plasma t_1/2_ of 5.09 h (Figure 8). More importantly, diblock P-THPs revealed a further increased circulation time, for which the plasma t_1/2_ of D-ss-THP, r-D-ss-THP, and D-_PVGLIGK_-THP were 6.95 h, 5.93 h, and 6.34 h, respectively (Figure 8). The increase in plasma t_1/2_ is mostly attributed to the larger sizes of diblock P-THPs as described above.

### 3.9. In Vivo Antitumor Effects of P-THP Conjugates

Finally, we examined and compared the in vivo antitumor effects of different P-THPs in a mouse sarcoma S180 solid tumor model. As shown in Figure 6, at a dose of 5 mg/kg (THP equivalent), L-THP significantly suppressed tumor growth up to 10 days after treatment, after which the tumor regrew rapidly, whereas all diblocks showed a much more remarkable tumor inhibitory effect than L-THP, which no apparent regrowth of tumors were observed up to 30 days after treatment (Figure 9A). In addition, no apparent decrease in the body weight of the mice was observed in all treatment groups (Figure 9B).

## 4. Discussion

In this study, we aimed to design and develop tumor-microenvironment-sensitive polymer nanomedicines with enhanced pharmacokinetics of the carried drug, THP and pH-sensitive THP release, and reactivation in tumor tissue. See Figure 10 for overall description of the diblock polymer design and activity principle. Generally, nanomedicines were accumulated in solid tumor tissues by employing the EPR effect-based accumulation. Such an effect could be enhanced by the increased size of the nanomedicines. Nevertheless, the degradability of such nanomedicines should be taken into consideration to avoid any unwanted accumulation of the carrier system after fulfilling its role in the organism. Here, two stimuli of biodegradability were selected to design the effective polymer-based nanomedicines with reduction- or matrix-metalloproteinase-driven degradation of the HMW diblock polymer structure. The developed biodegradable diblock polymer systems were designed for prolonged blood circulation and increased tumor accumulation combined with a tumor-microenvironment-sensitive intra- or extracellular degradation of the system, causing the drug activation and degradation of the carrier in in situ in tumorous tissue. Moreover, a simplified one step synthesis of highly effective polymer nanomedicines was employed, enabling the easy synthesis and potential scale-up of polymer nanomedicines.

The diblocks were successfully synthesized by the utilization of two different synthetic approaches. Either condensation of two linear polymers prepared by free radical polymerization or direct controlled RAFT polymerization using the bifunctional CTA were employed. Here, the RAFT polymerization enables the effective and elegant synthetic procedure for direct diblock synthesis. Such synthesis decreased the number of the reactive steps and enabled the final physico-chemical properties of the given diblock biodegradable polymer to be easily set up.

Moreover, polymer–THP conjugates proved their applicability as pH-sensitive polymer nanomedicines. The release from all the polymer–THP conjugates was highly accelerated with decreasing pH, thus showing the pH-sensitive behavior of the systems. We presume that the system will almost be stable during circulation in blood stream, and afterwards, the enhanced EPR effect-driven accumulation the THP will be reactivated by the release in mild acidic conditions of the tumor microenvironment. In addition, we verified the tumor-stimuli driven degradation of diblock polymers. Both the reductive environment and presence of the MMP-2 or MMP-9 led to the degradation of diblocks to polymer chains with size enabling their excretion from the body. Thus, we assume that nanomedicines could be degraded by the reductive or enzymatic degradation, and consequently eliminated from the body. Importantly, the degradation of diblocks by MMP could also lead to a deeper extravasation of diblock polymers into the tumorous tissue. We are convinced that the degradation of diblock, once accumulated in a solid tumor, could lead to the deeper penetration of a smaller linear polymer within the tumor mass. This notion was supported by the experiment using tumor cell spheroids. Despite having the larger sizes than L-THP, diblocks showed similar penetration profiles in spheroids compared to L-THP (Figure 7). Additionally, r-D-ss-THP even showed better and more rapid penetration than L-THP (Figure 7). These suggested that the rapid cleavage of S-S bonds in tumor tissue, and the distinctly sharp size distribution of RAFT polymer benefits the better penetration whereas the broad size distribution of other P-THP that contained larger sized P-THPs resulted in a slower penetration compared to RAFT polymer conjugates.

In parallel with the release profiles, diblocks showed similar in vitro cytotoxicity activities (Figure 5) and intracellular uptake (Figure 6). These findings suggested that biological activities (antitumor activities) of P-THPs were mostly due to the release of free THP, which supported the importance of the tumor pH-responsive bonds (hydrazone bond) of these P-THP conjugates. The critical difference between diblocks and L-THP is the size, which we anticipated in the superior pharmacokinetics of diblocks to L-THP. As expected, all diblocks that had a 2-fold larger hydrodynamic size than L-THP (Table 1), exhibited a more prolonged circulation time than L-THP (Figure 8). The increased plasma half-lives of diblocks ensured better tumor accumulation based on the EPR effect, as evidenced by the much-improved in vivo therapeutic effects compared to L-THP (Figure 9). In addition, by the treatment protocol in this study (i.e., bolus injection of 5 mg/kg), no apparent side effects (e.g., body weight loss) were observed in diblock polymer conjugates, indicating that the safety profiles were not significantly affected by the increased molecular sizes and prolonged circulation time of diblock polymer conjugates; however, further investigations are warranted to elucidate the safety, toxicity, and metabolic properties of diblock polymer conjugates.

## 5. Conclusions

Herein, we described the controlled synthesis and deep evaluation of biodegradable diblock polymer nanomedicines. A diblock polymer–THP conjugate structure was designed to fulfill all prerequisite criteria for effective nanomedicines in solid tumor treatments. An advanced synthetic procedure was introduced for diblock synthesis by employing a direct one-pot polymerization technique to create biodegradable polymer systems with suitable physico-chemical characterizations. Compared to conventional linear L-THP, diblock polymer conjugates showed a comparable cytotoxicity, intracellular uptake, and intracellular penetration properties in accordance with their release profiles, which are similar to that of L-THP. However, owing to their larger molecular sizes compared to L-THP, they exhibited a very prolonged circulation time, consequently resulting in superior in vivo antitumor activities compared to L-THP. These findings suggested the advantages of diblock polymer conjugates as anticancer polymer nanomedicine.

## Data Availability

The data presented in this study are available on request from the corresponding author.

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
