# Peer review of "Tumor Stimulus-Responsive Biodegradable Diblock Copolymer Conjugates as Efficient Anti-Cancer Nanomedicines"

_jpm, 2022, doi:10.3390/jpm12050698_

Round 1
Reviewer 1 Report
Manuscript “Tumor Stimulus-responsive Biodegradable Diblock Copolymer Conjugates as Efficient anti-cancer nanomedicines” described diblock polymer conjugate nanosystmes based on HPMA conjugated with THP drug as a novel (pH and enzyme)-tumor microenvironment responsive degradable polymer for cancer therapy. The data is not well-presented and it’s hard to follow what the authors mean. This can be improved by providing a graphical abstract. As well as writing a paragraph introducing the number of developed polymers, their main features and their abbreviations. Below are comments to the authors:
- It would be better to show the different steps (reactions) for synthesis of diblock copolymer in one figure.
- A schematic design showing working principle and novelty of diblock copolymer could be very helpful for readers.
- What was the rational for choosing drug pirarubicin (THP) and cell line C26?
- Data for THP characterization, including their size, shape, charge using DLS, FTIR, H-NMR, TEM should be provided.
- Authors indicated that reduction in polymer size due to SS bond reduction is key to improved penetration into cancer cells. This feature should be highlighted in the MS, especially in the abstract and introduction section.
- Although the designed diblock pHPMA nanosystems are tumor-specific responsive and biodegradable, how such drug conjugate can afford homogenous and deep penetration within the tumor region?
- Also, for in vivo application, reported by i.v injection, how the issue of protein corona formation can be avoided using polymeric conjugates?
Author Response
Manuscript “Tumor Stimulus-responsive Biodegradable Diblock Copolymer Conjugates as Efficient anti-cancer nanomedicines” described diblock polymer conjugate nanosystems based on HPMA conjugated with THP drug as a novel (pH and enzyme)-tumor microenvironment responsive degradable polymer for cancer therapy. The data is not well-presented and it’s hard to follow what the authors mean. This can be improved by providing a graphical abstract. As well as writing a paragraph introducing the number of developed polymers, their main features and their abbreviations. Below are comments to the authors:
Response: We thank the reviewer for his comments. Based on the comments we have significantly improved the manuscript. We have added self-explaining graphical abstract and also schematic design showing the working principle into the main text. The text describing the developed polymers was added into the manuscript.
It would be better to show the different steps (reactions) for synthesis of diblock copolymer in one figure.
Response: We thank the reviewer for tis important comment. We have improved the schemes as follows. Scheme 1 is now describing the synthesis of the bis-CTA. Scheme 2 is now showing the synthesis of the diblock polymer precursor in one step. Both Schemes are listed in attached file.
A schematic design showing working principle and novelty of diblock copolymer could be very helpful for readers.
Response: We introduced new figure 10 describing the working principle into the manuscript, as well as the graphical abstract. We hope that the main idea of the manuscript is now clear for the readers.
What was the rational for choosing drug pirarubicin (THP) and cell line C26?
Response: We chose THP because compared to commonly-used doxorubicin, it shows much rapid and higher intracellular uptake and cytotoxicity (please refer to our previous paper, doi: 10.1021/acs.molpharmaceut.6b00697), i.e., compared to HPMA-DOX, HPMA-THP exhibited much stronger antitumor effect. And in this study, we used mouse colon cancer C26 cells which is a routinely used cell line in our lab, the aim of the study using C26 is to examine and compare the uptake and cytotoxicity of different P-THP, we did not focus on the specific type of tumor in this study, C26 was used just as a representative of tumor cells. Thank you.
Data for THP characterization, including their size, shape, charge using DLS, FTIR, H-NMR, TEM should be provided.
Response: Within the manuscript the proper characterisation of the prepared polymer-THP systems was mentioned. The molecular weight, dispersity and hydrodynamic size in aqueous solution was determined, described and discussed. The developed DDS are water-soluble polymer systems, which form in the solution random coil, which can be effectively described by the hydrodynamic size, as could be seen in table 1. Zeta potential could not be determined easily, as the polymer system are not forming particles and thus the classic model of the zeta potential calculation can not be used. Moreover, the pHPMA systems are non-charged polymer systems, thus we believe that it is not worth of investigation. The TEM experiments were not carried out as the polymer systems have very low contrast as they are water-soluble and not forming any particular shape, thus we did not see effective to perform such analysis. The NMR analysis was performed for the monomers and low-molecular weight compounds. In the case of the polymer systems, the NMR do not help in the characterisation, as all the characteristics are precisely determined by other methods as described in the paper.
Authors indicated that reduction in polymer size due to SS bond reduction is key to improved penetration into cancer cells. This feature should be highlighted in the MS, especially in the abstract and introduction section.
Response: In this particular point we do not agree with the reviewer. We did not indicate that the SS bond reduction is the key for penetration. First, the SS reduction could be observed after the penetration into the cells by the glutathione and thioredoxins, thus the reduction occurs after the penetration not before. Importantly, we have found that the penetration is independent of the structure of liner or diblock polymer systems.
Although the designed diblock pHPMA nanosystems are tumor-specific responsive and biodegradable, how such drug conjugate can afford homogenous and deep penetration within the tumor region?
Response: The main aim of the diblock polymer development was the advancement of the pharmacokinetic of the polymer-drug systems. We have found that this aim was fully fulfilled. Importantly, we have found that the diblock polymer can penetrate into the tumor spheroid to the same extent as smaller linear monoblock polymers. Thus, we believe that the diblock polymer systems will be for the in vivo evaluation and treatment efficacy in solid tumor eradication much better than the linear monoblocks. We assume that deep penetration of the drug carried on the polymer system could be obtained by the combination of the penetration of diblock polymer, linear polymer after the degradation, drug released from the polymer system and also by the exosomes formed by the cell, to which the polymer-drug conjugate or free drug was already penetrated. In the moment, we cannot ensure the homogenous and deep drug penetration, as this investigation is out of the scope of present paper.
Also, for in vivo application, reported by i.v injection, how the issue of protein corona formation can be avoided using polymeric conjugates?
Response: We thank the reviewer for this highly important comment. The HPMA-based copolymers were recently recognised as highly non-fouling polymer materials and no soft or hard protein corona from most abundant proteins of human plasma was not found, ref. DOI: 10.1039/c7nr09355a. We have added the information about the non-fouling properties into the manuscript.

Reviewer 2 Report
The manuscript untitled: “Tumor Stimulus-responsive Biodegradable Diblock Copolymer Conjugates as Efficient anti-cancer nanomedici” is a very interesting and complete study about the preparation, physicochemical characterization, drug loading and release from stimuli-responsiveness polymers, and in vitro and in vivo studies. In my opinion, all the important parameters were evaluated in the study of the formulations.
Author Response
The manuscript untitled: “Tumor Stimulus-responsive Biodegradable Diblock Copolymer Conjugates as Efficient anti-cancer nanomedici” is a very interesting and complete study about the preparation, physicochemical characterization, drug loading and release from stimuli-responsiveness polymers, and in vitro and in vivo studies. In my opinion, all the important parameters were evaluated in the study of the formulations.
Response: We thank the reviewer for his comments and careful reading of the manuscript.
Round 2
Reviewer 1 Report
Manuscript can be accepted for publication.